# Assessment of the Emotional State of Parents of Children Starting the Vojta Therapy in the Context of the Physical Activity—A Pilot Study

**DOI:** 10.3390/ijerph191710691

**Published:** 2022-08-27

**Authors:** Kinga Strojek, Dorota Wójtowicz, Joanna Kowalska

**Affiliations:** Faculty of Physiotherapy, University of Health and Sport Sciences, 51-612 Wroclaw, Poland

**Keywords:** central coordination disorders, emotional state, anxiety, stress, mood, life satisfaction, Vojta therapy, stress coping strategies

## Abstract

The aim of the study was to assess the emotional state of parents at the moment of starting therapy for their children using the Vojta method in the context of the physical activity undertaken by the parents. The study involved 68 parents (37 mothers and 31 fathers) of children with central coordination disorders (CCD) presenting for consultation and therapy using the Vojta method. The authors’ questionnaires, the Perceived Stress Scale (PSS-10), the State-Trait Anxiety Inventory (STAI), the Patient Health Questionnaire (PHQ-9), the Satisfaction With Life Scale (SWLS), and the Inventory to Measure Coping Strategies with Stress (Mini-COPE) were used. As many as 84% of mothers and 77% of fathers presented high level of perceived stress. Comparative analysis showed a statistically significant difference in anxiety and life satisfaction between the groups of mothers and fathers studied. Taking declared physical activity into account, there was a statistically significant difference in stress and anxiety in the mothers’ group and a statistically significant difference in mood and life satisfaction in the fathers’ group. Promoting physical activity among parents of children with CCD can be helpful in maintaining better psycho-physical conditions and can also be a good tool in combating stress in difficult situations, such as the illness and therapy of a child.

## 1. Introduction

The illness or disability and therapy of a child changes the functioning of the whole family structure, is a multifaceted experience, and causes feelings of helplessness and uncertainty. It involves prolonged stress, anxiety, guilt and the need to reorganise an individual’s life to date. Many reports focus on the negative effects of care, therapy and treatment, as well as the barriers and limitations resulting from difficult situations such as the illness of a child [1,2].

Factors that exacerbate stress for parents include the specificity of the disorder and the degree of the child’s dysfunction. The more severe the disability and degree of dysfunction, the greater the parents’ anxiety about their child’s future and independence [3]. Researchers point out that parents of children with developmental disorders are exposed to physical and emotional burdens and may develop anxiety, burnout and depressive disorders [4,5,6].

Findings indicate an increased risk of depressive disorders in mothers of children with disabilities than in mothers raising children without developmental disorders [7,8]. In contrast, Dhungel et al. emphasised that fathers of disabled children had a higher prevalence of psychological distress and poor subjective health status than fathers of non-disabled children [9].

Parents’ attitudes towards the proposed therapy and treatment methods [10,11,12] and the need for comprehensive therapy appropriate to the child’s condition are among the many stress factors [13]. Modern physiotherapy offers many options for working with the child [14]. One of the leading methods used in stimulation is the neurokinesiological diagnostic and therapeutic concept developed by Vaclav Vojta. It is a method widely used in paediatric, adolescent and adult physiotherapy to promote normal movement patterns and eliminate pathological solutions [15]. The specific nature of working with children involves the child’s opposition, often involving screaming and the discomfort felt during therapy [4,16]. This has a significant impact on the feelings and attitudes of parents performing therapy [13,16]. The effectiveness of this method has been repeatedly confirmed [5,17,18,19,20,21]. Despite many studies indicating the high efficacy and positive effects of the therapy and parents’ increasing knowledge of it [4,22,23], the method still arouses a lot of emotion and concern [4,24]. Additionally, Kiebzak et al. showed that Vojta stimulation increased the level of free cortisol in saliva, but only 8.75% of the surveyed children exceeded the normative level. The authors emphasised that after Vojta’s intervention, the level of free cortisol in saliva decreased significantly, reaching the reference values after 20 min [16].

The emotional state of parents plays a key role in the rehabilitation process of children with psychomotor developmental disorders. Improving the child’s functioning is significantly influenced by parental involvement and support, responding to the child’s needs and adapting requirements to the child’s needs and abilities [4]. Parents have a daunting task of fulfilling caregiving tasks and rehabilitation while striving for control of their child’s symptoms and maintaining family balance [25]. Deterioration of a carer’s psycho-physical state adversely affects the quality of care provided [12,26].

Physical activity has an important impact on the emotional states of parents. Previous research confirms a significant link between the occurrence of nervous tension, depressive and anxious states, and regular physical activity [27,28,29]. Regular physical exercise reduces the risk of depressive symptoms and has a positive impact on quality of life [30,31]. In a study among women, an interaction between increased physical activity and a reduction in anxiety symptoms was demonstrated [29]. A difficult situation such as the diagnosis of abnormalities in a child and the need to introduce therapy can generate high level of stress, anxiety and worse mood in parents. There are few publications on the assessment of the emotional state of parents starting therapy for their children using the Vojta method and the role of physical activity in tackling the emerging stress and worse mood in this group of people. Hence, the aim of this study was to assess the emotional state of parents at the moment of starting the therapy of their children using the Vojta method, in the context of the physical activity undertaken by the parents and to answer the study questions: (1) What is the emotional state of mothers and fathers at the start of Vojta therapy; and (2) Is there a difference in the level of stress and anxiety, mood and life satisfaction in physically active and physically inactive parents?

The results of these pilot studies may have practical implications. They will also be used to conduct the main studies. Knowing the emotional state of parents prior to the child’s therapy using the Vojta method may lead to changes in physiotherapy treatment patterns by paying attention not only to the child but also to the parent-therapist. The parent-therapist also needs support and incentive to take care of their well-being through, for example, regular physical activity, which can play a predictive role.

## 2. Materials and Methods

### 2.1. Participants and Procedure

The pilot study was conducted in Wroclaw from November 2019 to April 2020. The research was interrupted due to the COVID-19 pandemic. The study group consisted of parents of children with central coordination disorders (CCD) who, in accordance with medical advice, attended a consultation with a Vojta method therapist and were subsequently qualified for therapy of their child using this method. The study parents fulfilled the following inclusion criteria: consent to participate in the study, parents of children aged 0–2 years, willingness to start their child’s therapy with the Vojta method, and no previously diagnosed or pharmacologically treated serious psychiatric disorders (e.g., depression).

During the first two years after the baby is born, central coordination disorders (CCD) are most often diagnosed. It is a temporary diagnosis and parents with a child are referred to therapy, for example, using the Vojta method.

The parents were informed of the principles and objectives of the study and gave their voluntary (written) consent for their participation. The study received approval from the Senate Commission for the Ethics of Scientific Research of the University of Health and Sport Sciences in Wroclaw (reference no. 41/2019). This study was conducted in accordance with the Helsinki Declaration.

The inclusion criteria were met by 74 parents, but 6 of them (fathers) did not complete all the questionnaires. Only those questionnaires that were correctly and completely filled out were included in the analysis. Therefore, the final study group consisted of 68 parents, including 37 mothers and 31 fathers (31 couples and 6 mothers), aged 31.6 (±4.3). They were the parents of 34 children (17 boys and 17 girls; 26 children were the first child in the family and 8 were the second child in the family; 7 children with severe CCD and 27 with moderate CCD), the mean age 3.6 (±2.0) months (range 1.5–9.0 months). The characteristics of the study group are shown in Table 1.

### 2.2. Measure Tools

The Perceived Stress Scale (PSS-10), the State-Trait Anxiety Inventory (STAI), the Patient Health Questionnaire (PHQ-9), the Satisfaction with Life Scale (SWLS), and the Inventory to Measure Coping Strategies with Stress (Mini-COPE) were used. Sociodemographic data and information about parents’ knowledge, their physical activity and concerns about Vojta therapy were collected using the authors’ own questionnaire.

The PSS-10 was created by Cohen et al., a Polish adaptation from Juczyński and Ogińska-Bulik. The questionnaire consists of 10 questions that measure the subjective perceived stress associated with stressful situations. The higher the score (max. 40 points), the higher the intensity of perceived stress. The raw score is converted into sten: 1–4 sten (0–13 points) is considered low, 5–6 sten (14–19 points)—moderate, and 7–10 (20–40 points) is considered a high level of stress. Cronbach’s alpha was 0.86 [32].

The STAI is used to diagnose the severity of anxiety as a state (STAI X-1) and anxiety as a personality trait (STAI X-2). The questionnaire consisted of 40 responses. The maximum score is 80 points. The obtained score indicates the level of anxiety, where the higher the score, the higher the level of anxiety. The high level of anxiety for the STAI (X1) was a score above 44, and for the STAI (X2), a score above 46. The psychometric properties of the Polish version are similar to the original (Cronbach’s alpha is 0.89 for STAI X-1 and 0.83 for STAI X-2) [33].

The PHQ-9 is a screening tool for diagnosing depression. The Polish language version consists of 9 questions each on the presence of depressive symptoms and 1 additional question. Depending on the severity of a given symptom for the last two weeks, the respondent marks one of the answers 0–3, where 3 indicates the symptoms that occurred most frequently. A total of 27 points can be scored for the entire survey. Scores of 5, 10, 15 and 20 represent cutpoints for mild, moderate, moderately severe and severe depression, respectively. The PHQ-9 is currently one of the best tools for diagnosing depression in people aged 18 to 60 years. Cronbach’s alpha was 0.88 [34].

The SWLS measures the subject’s subjective sense of life satisfaction. The higher the score, the more satisfied the participant is with life. This tool uses raw results of the Polish standards: 5–17 points, low satisfaction; 18–23 points, average satisfaction; 24–35 points, high satisfaction. The psychometric properties of the Polish version are satisfactory and similar to the original. Cronbach’s alpha was 0.81 [35].

The Mini-COPE inventory developed by Carver in Polish adaptation by Juczyński and Ogińska-Bulik, is used to evaluate typical coping strategies and reactions in situations where the individual experiences severe stress. It consists of 28 statements that refer to 14 strategies that describe various coping styles. The higher the score, the more frequently the given strategy is employed by the respondent. Strategies such as active coping, planning and use of instrumental support are referred to as “problem-focused.” Use of emotional support, religion or denial are considered “emotion focused.” Venting, self-distraction, behavioural disengagement, substance use and a sense of humour signify avoidance; nevertheless, they can bring short-term relief. Split-half reliability was 0.86 (Guttman split-half coefficient—0.87) [32].

To reduce the effect of the personality of the physiotherapist and his/her ability to explain the therapy, all parents surveyed had a scheduled appointment with the same physiotherapist (a certified therapist in the Vojta method with many years of work experience).

### 2.3. Data Analysis

The normality of distribution has not been confirmed (the Shapiro–Wilk test). In order to describe the results, a median and inter-quartile range (IQR) and, additionally, a mean and standard deviation (SD) were used. The Mann–Whitney U test was used to assess the significance of differences between the groups. The χ^2^ test was used to verify the significance of differences between groups in terms of their selected sociological traits. The significance level was assumed at *p* < 0.05. Additionally, to determine the quantity of the effect of differences between the examined groups, a corrected Cohen’s d test was used. Cramer’s V coefficient was used to calculate the effect size of χ^2^ test.

## 3. Results

The compared groups of mothers and fathers did not differ significantly in terms of age, place of residence, education, economic status and declared physical activity (Table 1).

The entire study group (*n* = 68) was found to have high level of perceived stress (mean 22.9 ± 3.6; median 23.0) and anxiety (46.3 ± 3.8 as a state; median 46.0). As many as 84% of the surveyed mothers and 77% of the surveyed fathers presented high levels of perceived stress. Mood disorders were found in 65% of mothers and 52% of fathers. The most common concerns of parents of children in whom the Vojta method was to be introduced were concerns about their child’s health, their child’s future and the correct administration of the therapy.

Comparative analysis showed a statistically significant difference in trait anxiety and life satisfaction between the mother and father groups studied (Table 2).

A significantly higher level of trait anxiety and a significantly worse mood, as well as significantly better life satisfaction, were observed in the group of younger parents (aged 30 and under) compared to the group of parents aged 30 and over (*p* = 0.0349, *p* = 0.0139, *p* = 0.0193, respectively).

Taking into account the declared physical activity across the study group, there was a statistically significant difference in stress and life satisfaction (Table 3).

A statistically significant difference was observed in the level of stress and anxiety in the group of mothers according to whether physical activity was undertaken or not. However, in the group of fathers, a statistically significant difference was found for mood and life satisfaction (Table 4).

The most frequently chosen strategies for coping with stress in the whole study group (*n* = 68) were Active coping, Planning and Acceptance.

Women were significantly more likely to choose strategies such as Running to religion and Seeking instrumental support, compared to the group of male respondents (Table 5).

Physically inactive people were significantly more likely to choose the strategy: Behavioural disengagement, compared to the physically active group (*p* = 0.0307). There were no statistically significant differences in the other stress-coping strategies.

## 4. Discussion

Parents play a key role in Vojta therapy and have a great deal of responsibility for the developmental progress of their children. The introduction of early physiotherapy intervention in a child is a challenge for parents and involves many physical and psychological difficulties [4]. It is a highly stressful situation and contributes to mood disorders. Szałowska et al. emphasised that high level of perceived stress are more likely to be present prior to treatment [36]. The present study showed high levels of stress in the entire study group, with more than half of the subjects reporting depressive symptoms. These findings are in line with reports by other authors [37,38,39], although many highlight a significant difference by sex, noting that higher level of stress and incidence of mood disorders are significantly more frequently observed in the group of mothers than in the group of fathers [38,40]. The cultural factors of the country in question and the perception of the father’s role in the modern family are probably of great importance here. According to Waligóra, it amounts to providing economic security for the family and is characterised by a lower emotional bond to the child than that of the mother [41]. Often, caring for a child with a disability is considered the mother’s responsibility [42]. They are therefore the ones who bear a lot of the responsibility for a sick child and are more likely to experience difficulties, stress and the emergence of mood disorders [38,39]. Other researchers point out that the joint involvement of both parents allows for the strengthening of the parent–child bond and a closer partner relationship, which is an important aspect in ongoing therapy [43,44].

Although the stress levels and declared concerns of mothers and fathers (about the child’s health, the child’s future, and the correct administration of therapy) were similar, the mothers’ group had significantly higher levels of anxiety. Other researchers have also confirmed that the level of perceived anxiety for a family with a child with a chronic illness is higher in the mother’s group than in the father’s group [45,46].

Nevertheless, the level of satisfaction and life satisfaction was significantly higher in the mothers than in the fathers, in contrast to a study by Basińska and Wędzińska, in which the fathers of children with cerebral palsy had higher levels of life satisfaction than the mothers [47]. Caring for a sick or disabled person can have a negative impact on the carer, but it can also give a sense of satisfaction, a sense of duty towards loved ones, a sense of control over the situation and a special meaning to the effort invested [25].

Higher level of trait anxiety and worse mood, but higher levels of life satisfaction were associated with younger parents (≤30). The arrival of a sick child in the family often leads to adjustments in life plans. Such decisions are always more difficult for young people and can generate more stress. This may result from a lower parental sense of competence, which is more common for younger parents. Moreover, younger parents are much more likely than older parents to use problem-focused coping in general situations and those connected with the caregiving and upbringing of a child (planning strategy and use of instrumental support [48].

Similar strategies (active coping and planning) were chosen most frequently, both in the whole sample of parents, as well as among mothers and fathers. Such strategies belong to the group of problem-focused strategies. In a study by Aftyka et al., active coping and planning were also the most frequently chosen strategies by parents [49]. Such proactive strategies are chosen when it is foreseeable that the situation will improve. Parents starting Vojta therapy expect to see an improvement in their child’s functional status, so they may be using action-based strategies. Research by other authors shows that mothers of sick children most often use a strategy based on avoiding confrontation with the stressful situation, reducing unpleasant emotional tension and focusing on their own emotions [50].

Byra and Parchomiuk’s study showed that mothers chose to cope with difficulties by accepting, resolving difficulties, and seeking emotional support [51].

It should be noted that in the present study, mothers were significantly more likely to choose the strategies: Seeking instrumental support and Running to religion. This is supported by the findings of Aftyka et al., who emphasised that women are not dominated by just one group of strategies and are significantly more likely than men to choose the strategy seeking instrumental support [49]. Such results indicate that the mothers surveyed attempted to confront and resolve difficult situations in addition to taking an emotional approach.

Taking physical activity into account, one can see its positive impact on the emotional state of the parents surveyed. Parents who declared regular physical activity had significantly lower stress level and significantly higher life satisfaction. Mothers who were physically inactive had significantly higher level of stress and anxiety. In contrast, the group of physically inactive fathers reported a significantly poorer mood and lower life satisfaction. These findings suggest that undertaking regular physical activity can be an effective means (also preventive) for both parents to combat stress and anxiety and maintain wellbeing in difficult situations, such as the illness and therapy of a child. Previous reports indicate that physical activity can be highly beneficial in preventing the onset of depressive symptoms at any age [27,28,30,52].

Research by Herring et al. highlights the existing relationship between physical activity and the risk of anxiety [29]. For example, exercise, yoga, meditation, tai chi, or qi gong can improve the symptoms of depression and anxiety disorders [53]. Physical activity produces its antidepressant effect through multiple biological and psychosocial pathways [28].

A disturbing and thought-provoking result seems to be that parents who were physically inactive were the most likely to choose the Behavioural disengagement coping strategy for stress, i.e., these individuals were significantly more likely to give up on goal attainment than those who were physically active. This strategy is not very effective and may provide only short-term relief [32]. Physically inactive parents were also characterised by higher level of stress, and according to the results of Repka et al., the higher the level of stress and fatigue, the more frequent the choice of precisely such Behavioural disengagement strategies [54].

In summary, associations between physical activity and depression suggest significant mental health benefits from being physically active [27]. Therefore, parents of children in whom the Vojta method is introduced should be encouraged to increase their physical activity to improve their mental health. In addition, parents should have psychological support and access to knowledge on how to cope with stress and which strategies to use to successfully overcome stress. Therefore, there is a need to carefully analyse the mental and physical conditions of the parents in choosing the right therapy for their child so that they can correctly implement a physiotherapy programme to ensure that their child’s condition improves, especially when starting therapy.

The present study has a few limitations. A main limitation of the survey is the non-random sampling; thus, the results obtained should not be generalised. The tests performed are screening in nature and the results are not equivalent to making a diagnosis. The selected tools have satisfactory psychometric properties but are more subjective and can create a bias risk. Physical activity was assessed based only on parents’ responses to the question regarding regular physical activity. To strengthen the findings, the size of the study group should be increased. The study was carried out at a single point in time, i.e., at the beginning of Vojta therapy, which is a clear limitation, as it does not show the dynamics of changes in the emotional state of the parents surveyed. In future research, incorporating the measurement of stress biomarkers should be considered. The study also did not consider the severity of the disorders present in the children, only their ages. In addition, the study group consisted of parents who had made the decision to enter therapy and take on the role of parent therapist. Their motivation may be high, which may involve active coping strategies to deal with stress.

## 5. Conclusions

The results obtained indicate that there is a difference in the emotional states of mothers and fathers of children with CCD, especially when a specialised rehabilitation method, such as the Vojta method, is introduced. Thus, in a comprehensive rehabilitation model, attention should be paid not only to the child but also to the parents.

Educational activities and the promotion of physical activity among parents of children with CCD, who require rehabilitation using the Vojta method can be helpful in maintaining better mental and physical condition and may also be a good tool to combat stress in difficult situations such as the illness and therapy of a child.

## Figures and Tables

**Table 1 ijerph-19-10691-t001:** Characteristics of the two study groups (the Mann–Whitney U test and χ^2^ test).

	All Group *n* = 68	Mathers *n* = 37	Fathers *n* = 31	*p*	*Effect Size*
					*Cohen’s d*
Age:					
Mean (SD)	31.6 (4.3)	30.5 (6.1)	32.9 (4.6)		
Median (IQR)	32.0 (5.0)	31.0 (5.0)	33.0 (5.0)	0.1443	0.4388
Range	23–43	23–40	24–43		
	*n* (%)			*p* (χ^2^)	* **Cramer’s V** *
Place of residence:					
Village	5 (7)	3 (8)	2 (6.5)	0.7943	0.03
City/Town	63 (93)	34 (92)	29 (93.5)		
Education:					
Secondary	13 (19)	5 (14)	8 (26)	0.1991	0.15
High	55 (81)	32 (86)	23 (74)		
Economic situation:					
Working	63 (93)	32 (86)	30 (97)	0.1261	0.18
Not working	5 (7)	5 (14)	1 (3)		
Physical activity (at least twice a week/min. 120 min)					
Yes	41 (60)	24 (65)	17 (55)	0.4000	0.10
No	27 (40)	13 (35)	14 (45)		

**Table 2 ijerph-19-10691-t002:** Results of the studied parameters in the whole study group and according to sex (Mann–Whitney U test).

Scales	All, *n* = 68		Mathers, *n* = 37		Fathers, *n* = 31		*p*	*Effect Size*
	Mean (SD)	Median (IQR)	Mean (SD)	Median (IQR)	Mean (SD)	Median (IQR)		*Cohen’s d*
PSS-10	22.9 3.6	23.0 5.0	23.1 3.5	23.0 4.0	22.7 3.7	23.0 6.0	0.6818	0.11
STAI (X-1)	46.3 3.8	47.0 5.0	46.4 3.6	47.0 4.5	46.3 4.1	46.0 5.0	0.7948	0.02
STAI (X-2)	45.2 5.1	44.0 5.0	46.9 4.7	46.0 6.5	43.2 4.9	43.0 4.0	0.0009 *	0.77
PHQ-9	6.2 4.5	5.0 6.0	6.5 4.2	5.0 6.0	5.7 4.9	5.0 7.0	0.2846	0.17
SWLS	24.2 5.6	25.0 7.75	25.3 5.6	26.0 7.0	22.7 5.3	24.0 6.0	0.0214 *	0.48

PSS-10, the Perceived Stress Scale; STAI, the State-Trait Anxiety Inventory; PHQ-9,the Patient Health Questionnaire; SWLS, the Satisfaction With Life Scale; IQR, inter-quartile range; SD, standard deviation; * *p* < 0.05.

**Table 3 ijerph-19-10691-t003:** Results of selected scales according to declared physical activity in the whole study group, (Mann–Whitney U test).

Scales	Physical Activity					
	NO; *n* = 27		YES; *n* = 41		*p*	*Effect Size*
	Mean (SD)	Median (IQR)	Mean (SD)	Median (IQR)		*Cohen’s d*
PSS-10	24.0 3.7	24.0 5.0	22.2 3.3	22.0 4.5	0.0315 *	0.44
STAI (X-1)	46.3 4.5	47.0 5.0	46.4 3.3	46.0 5.0	0.8258	0.02
STAI (X-2)	46.2 5.7	45.0 9.0	44.5 4.5	44.0 5.0	0.2149	0.33
PHQ-9	7.5 5.1	6.0 8.0	5.3 3.8	5.0 5.0	0.0784	0.49
SWLS	22.8 5.1	23.5 6.25	25.3 5.2	26.0 6.5	0.0324 *	0.48

PSS-10, the Perceived Stress Scale; STAI, the State-Trait Anxiety Inventory; PHQ-9,the Patient Health Questionnaire; SWLS, the Satisfaction With Life Scale; IQR, inter-quartile range; SD, standard deviation; * *p* < 0.05.

**Table 4 ijerph-19-10691-t004:** Results of selected scales in relation to declared physical activity in the groups of mothers and fathers surveyed (Mann–Whitney U test).

Scales	Physical Activity-Mothers *n* = 37					*Effect* *Size*	Physical Activity–Fathers *n* = 31					*Effect* *Size*
	YES; *n* = 24		NO; *n* = 13		*p*		YES; *n* = 17		NO; *n* = 14		*p*	
	Mean (SD)	Median (IQR)	Mean (SD)	Median (IQR)		*Cohen’s d*	Mean (SD)	Median (IQR)	Mean (SD)	Median (IQR)		*Cohen’s d*
PSS-10	22.3 3.4	22.0 4.5	24.7 3.2	24.0 4.0	0.0227 *	0.73	22.1 3.2	21.0 5.5	23.4 4.2	24.5 7.5	0.1711	0.26
STAI (X-1)	45.8 3.1	46.0 6.0	47.1 4.3	48.0 5.5	0.1292	0.35	46.9 3.6	48.0 5.5	45.5 4.6	45.5 5.25	0.2266	0.34
STAI (X-2)	45.8 4.8	46.0 5.0	48.8 4.1	49.0 8.0	0.0322 *	0.67	42.6 3.7	43.0 4.5	43.9 6.1	42.5 6.25	0.4286	0.26
PHQ-9	6.3 4.0	5.0 5.0	6.8 4.5	5.0 7.5	0.4364	0.12	3.8 3.0	3.0 5.0	8.1 5.7	7.5 9.25	0.0107 *	0.94
SWLS	26.1 5.4	26.5 6.75	23.9 5.8	25.0 7.5	0.3173	0.39	24.3 4.9	24.3 7.5	20.7 5.3	22.0 6.75	0.0455 *	0.70

PSS-10, the Perceived Stress Scale; STAI, the State-Trait Anxiety Inventory; PHQ-9,the Patient Health Questionnaire; SWLS, the Satisfaction With Life Scale; IQR, inter-quartile range; SD, standard deviation; * *p* < 0.05.

**Table 5 ijerph-19-10691-t005:** Strategies for coping with stress according to sex (Mann–Whitney U test).

Mini COPE	All Study Group, *n* = 68	Mothers *n* = 37		Fathers *n* = 31		*p*	*Effect Size*
	Mean (SD)	Mean (SD)	Median (IQR)	Mean (SD)	Median (IQR)		*Cohen’s d*
1. Active coping	2.8 * (0.7)	2.8 * (0.6)	3.0 (1.0)	2.9 * (0.7)	2.0 (3.0)	0.4325	0.15
2. Planning	2.8 * (0.6)	2.7 * (0.6)	2.0 (1.0)	2.8 * (0.6)	2.0 (3.0)	0.4602	0.17
3. Positive reframing	2.6 (0.8)	2.6 (0.8)	2.0 (4.0)	2.7 * (0.7)	2.0 (4.0)	0.5000	0.13
4. Acceptance	2.7 * (0.7)	2.6 (0.7)	2.0 (4.0)	2.7 * (0.7)	2.0 (4.0)	0.4721	0.14
5. Sense of humour	1.8 (0.5)	1.8 (0.5)	1.0 (5.0)	1.8 (0.6)	1.0 (5.0)	0.2327	0.0000
6. Running to religion	1.6 (1.1)	2.0 (1.1)	1.0 (4.0)	1.1 (0.9)	0.0 (1.2)	0.0202 **	0.89
7. Seeking emotional support	2.5 (0.8)	2.6 (0.7)	2.0 (1.0)	2.5 (0.8)	2.0 (4.0)	0.1711	0.13
8. Seeking instrumental support	2.4 (0.6)	2.7 * (0.7)	2.0 (1.0)	2.0 (0.7)	2.0 (1.0)	0.0048 **	1.00
9. Self-distraction	2.5 (0.7)	2.6 (0.7)	2.0 (4.0)	2.4 (0.7)	2.0 (4.0)	0.2420	0.28
10. Denial	1.1 (0.7)	1.0 (0.8)	0.0 (1.0)	1.1 (0.6)	0.0 (1.0)	0.4920	0.14
11. Venting	2.4 (0.6)	2.4 (0.5)	1.5 (4.0)	2.3 (0.6)	1.0 (4.0)	0.2270	0.18
12. Substance use	0.7 (0.6)	0.6 (0.5)	0.0 (0.0)	1.0 (0.7)	0.0 (1.0)	0.1251	0.66
13. Behavioural disengagement	1.7 (0.7)	1.7 (0.8)	1.0 (5.0)	1.6 (0.7)	1.0 (5.0)	0.3632	0.13
14. Self-blame	2.1 (0.8)	2.2 (0.7)	1.0 (4.0)	1.9 (0.8)	1.0 (2.0)	0.2514	0.40

IQR, inter-quartile range; SD, standard deviation; * The most commonly employed strategies; ** *p* < 0.05.

## Data Availability

The data that support the findings of this study are available from the corresponding author upon request.

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
