# Peer review of "Assessment of the Emotional State of Parents of Children Starting the Vojta Therapy in the Context of the Physical Activity—A Pilot Study"

_ijerph, 2022, doi:10.3390/ijerph191710691_

Round 1

Reviewer 1 Report

Information about the experimental group is missing:
- the age of the child (newborn, infant, toddler) and the order of the child in the family - neither is specified
- parents of how many children participated in the research (it was always a complete pair);
- was it addressed whether rehabilitation exercises are provided by both parents - given the need for regular exercise - e.g. what is the ratio of exercises provided by the mother and father (will it affect stress levels)
- the text mentions "disabled child" but the indication for each patient is not specified - the introduction talks about central coordination disorder, which is only a unit indicating risks in the child's development but is not synonymous with disability;
- what level of regular physical activity was taken as positive (hours, performance, ...?)?
- the personality of the physiotherapist and his/her ability to explain the importance of the therapy carried out and to show its effect - who carried out the physiotherapy?
Without knowing this information, it is not relevant to compare parental behaviour.
What are the results of the tests and scores used in parents of infants without indicated therapy? Has it been published?

Author Response

Dear Reviewer,

We are grateful for the positive and constructive comments that originated in the review process. We have carefully reviewed the comments and have revised the manuscript accordingly. Our  responses are given in a point-by-point manner below.

Our manuscript has been reviewed by a native speaker and we have a cerificate.

Yours faithfully,
Authors

Reviewer 2 Report

Dear Authors,

congratulations on your interesting approach to the topic of the Vojta method. When preparing the work for publication, please take into account the following comments:

1. line 27 - please delete the duplicate "The",

2. lines 77 and 78 - please use the phrase central coordination disorder, without the word "nervous" in accordance with the terminology of the ZCK,

3. table 1 - please consider whether you need the information contained in table 1: place of residence, education and economic situation, because they are not analyzed in the calculation of the results, 

4. Quoting paper no 16, it should be emphasized in the discussion that after Vojta's intervention, the level of free cortisol in saliva decreased significantly, reaching the reference values after 20 minutes, and Vojta stimulation increased the level of free cortisol in saliva, but only 8.75% of the surveyed children exceeded the normative level.

Author Response

(The authors gave the same response as above.)

Author Response

(The authors gave the same response as above.)

Round 2

Author Response

Dear Reviewer,
Thank you for any comments regarding our work. Detailed answers are provided in the attachment.
